# Diffuse and Disperse: Image Generation with Representation Regularization

## Abstract

The development of diffusion-based generative models over the past decade has largely proceeded independently of progress in representation learning. These diffusion models typically rely on regression-based objectives and generally lack explicit regularization. In this work, we propose *Dispersive Loss*, a simple plug-and-play regularizer that effectively improves diffusion-based generative models. Our loss function encourages internal representations to disperse in the hidden space, analogous to contrastive self-supervised learning, with the key distinction that it requires no positive sample pairs and therefore does not interfere with the sampling process used for regression. Compared to the recent method of representation alignment (REPA), our approach is self-contained and minimalist, requiring no pre-training, no additional parameters, and no external data. We evaluate Dispersive Loss on the ImageNet dataset across a range of models and report consistent improvements over widely used and strong baselines. We hope our work will help bridge the gap between generative modeling and representation learning.

## 1 Introduction

Diffusion generative models (Sohl-Dickstein et al., 2015; Song & Ermon, 2019; Ho et al., 2020) have demonstrated remarkable performance in modeling complex data distributions, but their success remains largely disconnected from advances in representation learning. The training objectives of diffusion models typically consist of a *regression* term focused on reconstruction (*e.g.*, denoising), yet lack an explicit *regularization* term on the representations learned for generation. This paradigm for image *generation* stands in contrast to its counterpart in image *recognition*, where representation learning has been a core topic and a driving force over the past decade (Bengio et al., 2013).

In the field of representation learning, self-supervised learning has made significant progress in learning general-purpose representations applicable to a wide range of downstream tasks (*e.g.*, (He et al., 2020; Chen et al., 2020; He et al., 2022; Oquab et al., 2023)). Among these approaches, contrastive learning (Chopra et al., 2005; Hadsell et al., 2006; Oord et al., 2018; He et al., 2020; Chen et al., 2020) offers a conceptually simple yet effective framework for learning representations from sample pairs. Intuitively, these methods encourage attraction between sample pairs considered similar ("positive pairs") and repulsion between those considered dissimilar ("negative pairs"). Representation learning via contrastive learning has been shown useful across a variety of recognition tasks, including classification, detection, and segmentation. However, the effectiveness of these learning paradigms for generative modeling remains an underexplored problem.

In light of the potential of representation learning for generative modeling, Representation Alignment (REPA) (Yu et al., 2024) has been proposed to leverage the capabilities of pre-trained, off-the-shelf representation models. This method trains a generative model while encouraging alignment between its internal representations and external pre-trained representations. As a pioneering effort, REPA has revealed the importance of representation learning in the context of generative modeling; however, its instantiation depends on additional pre-training, extra model parameters, and access to external data. Developing a *self-contained* and *minimalist* approach to representation-based generative modeling is still an essential topic in this line of research. In this paper, we propose *Dispersive Loss*, a flexible and general plug-and-play regularizer that integrates self-supervised learning into diffusion-based generative models. Our core idea is simple: alongside the standard regression loss on the model output, we introduce an objective that regularizes the model's internal representations (Fig. 1). Intuitively, Dispersive Loss encourages internal representations to spread out in the hidden space, analogous to the repulsive effect

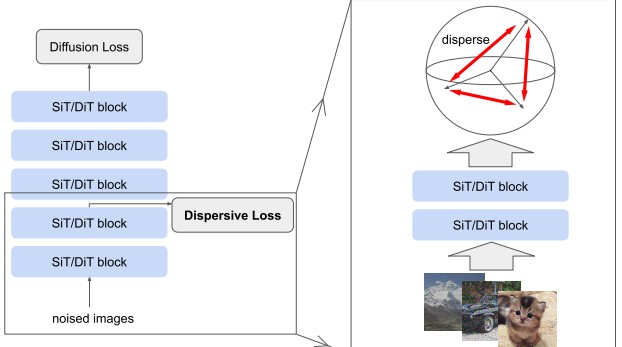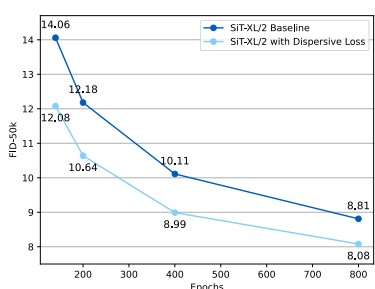

Figure 1: **Dispersive Loss for Generative Modeling**. **Left**: A standard diffusion-based model (*e.g.*, SiT (Ma et al., 2024) or DiT (Peebles & Xie, 2023)) with a regression-driven diffusion loss, augmented by a *Dispersive Loss* applied to an intermediate block. **Right**: A zoom-in on the first few blocks with Dispersive Loss. Our loss encourages intermediate representations to disperse in the hidden space. It operates on the same input batch of noised images, shares the existing network blocks and computations, introducing negligible overhead, no additional parameters, and no reliance on external data.

Figure 2: **Effectiveness of Dispersive Loss.** We show FID-50k without classifier-free guidance on ImageNet 256×256 at different training epochs, comparing the standard SiT-XL/2 baseline (Ma et al., 2024) with its counterpart trained with Dispersive Loss. More results and details are in Section 4.

in contrastive learning. Meanwhile, the original regression loss (*e.g.*, denoising) naturally serves as an alignment mechanism, eliminating the need to manually define positive pairs as in contrastive learning.

In a nutshell, Dispersive Loss behaves like a "contrastive loss *without* positive pairs"—and as such, unlike contrastive learning, it requires neither two-view sampling, specialized data augmentation, nor an additional encoder. The training pipeline can follow standard practice used in diffusion-based models (and their flow-based counterparts)[1], with the only distinction being an additional regularization loss that has negligible overhead. In comparison to the REPA mechanism, our method requires no pre-training, no additional model parameters, and no external data. With a self-contained and minimalist design, our method clearly demonstrates that representation learning can benefit generative modeling without relying on external sources of information.

We evaluate the effectiveness and generality of Dispersive Loss through extensive experiments. Our results show that Dispersive Loss consistently improves upon strong and widely used baselines in diffusion models, namely, DiT (Peebles & Xie, 2023) and SiT (Ma et al., 2024) (Fig. 2), across a wide range of model scales. We also find that various formulations of Dispersive Loss are consistently beneficial, demonstrating the robustness and generality of our approach; by comparison, incorporating a contrastive loss can degrade performance. Beyond multi-step diffusion models, we further apply Dispersive Loss to the recent MeanFlow framework (Geng et al., 2025), achieving state-of-the-art performance in *one-step* diffusion-based generation. We hope that the simplicity, effectiveness, and generality of Dispersive Loss will help bridge the gap between generative modeling and representation learning.

## 2 RELATED WORK

**Diffusion Models.** Diffusion probabilistic models (Sohl-Dickstein et al., 2015) formulate generative modeling as a progressive noising/denoising process, trained via a variational lower bound. This process is substantially simplified and improved by reformulating it as a denoising-based reconstruction objective (Ho et al., 2020). Subsequent work (Song et al.; Nichol & Dhariwal, 2021; Dhariwal & Nichol, 2021; Karras et al., 2022) further advances this direction by improving noise schedules and sampling strategies. Flow matching models (Lipman et al.; Liu et al.; Albergo & Vanden-Eijnden, 2023) extend the diffusion paradigm by focusing on velocity-based formulations and modeling deterministic trajectories. Despite these advancements, the regression-based training objective remains central to diffusion-based models and

---

[1]In the context of this work, we do not distinguish between diffusion methods and flow matching methods (Lipman et al.; Liu et al.; Albergo & Vanden-Eijnden, 2023), and refer to both under the umbrella term of "diffusion models".

their variants. ==InfoDiffusion== (Wang et al., 2023), ==DiffRep== (Mittal et al., 2023), ==and SODA== (Hudson et al., ==2024) demonstrate that diffusion objectives can yield strong self-supervised representations competitive with contrastive or masked modeling approaches. Other studies== (Zhang et al., 2022b; Yang & Wang, 2023) ==further show that representations can be effectively extracted from pre-trained diffusion models.==

**Self-Supervised Learning.** In parallel with advances in generative modeling, self-supervised representation learning has made steady progress in visual recognition. Contrastive learning (Chopra et al., 2005) has become a dominant self-supervised paradigm which encourages similarity between positive pairs and dissimilarity between negative pairs. Modern contrastive learning methods (He et al., 2020; Chen et al., 2020), driven by advances such as information-based contrastive losses (Oord et al., 2018) and large-scale negative sampling, have demonstrated encouraging results that can rival those of supervised representation learning. Interestingly, there is a specific line of research on "negative-free" contrastive learning (Grill et al., 2020; Chen & He, 2021), which lead to representation learning results that are competitively strong. Within this context, our method can be interpreted as "positive-free" contrastive learning, which is an underexplored direction in standard self-supervised learning.

Beyond contrastive learning, masked modeling (Devlin et al., 2019) has been established as another effective self-supervised learning paradigm in vision (He et al., 2022; Bao et al.; Zhang et al., 2022a). Compared to contrastive learning, masked modeling methods apply more destructive corruption to the input data than typical data augmentations. In contrast, our method aims to introduce no interference to the input samples for diffusion training.

**Representation Learning as Auxiliary Tasks.** Beyond the common pre-training/fine-tuning paradigm, representation learning can be performed as an auxiliary task jointly with the main objective. Supervised contrastive learning (Khosla et al., 2020) extends classical supervised models (*e.g.*, classification) with extra contrastive objectives. Self-supervised language-image pre-training (SLIP) (Mu et al., 2022) augments its contrastive counterpart (CLIP (Radford et al., 2021)) with a self-supervised objective trained in parallel.

In the context of image generation, REPA (Yu et al., 2024) explores a line of research on enhancing generative modeling with auxiliary representation learning. REPA aims to align the intermediate representations of a generative model to those from a frozen, high-capacity, pre-trained encoder, which may itself be trained using external data and diverse objectives. SARA (Chen et al., 2025) further advances this approach by introducing structural and adversarial representation alignment. In the multimodal setting, SoftREPA (Lee et al., 2025) extends REPA by aligning noisy image representations with soft text embeddings. While REPA and its extensions yield substantial gains in practice, they rely on additional pre-training overhead and, more notably, on *external sources of information*. It is difficult to disentangle whether the improvements from representation alignment arise from a self-supervised objective or primarily from increased compute and access to external data.

## 3 METHODOLOGY

### 3.1 DISPERSIVE LOSS

At the core of our method is the idea of regularizing a generative model's internal representations by encouraging their *dispersion* in the hidden space. We refer to the original regression loss in diffusion-based models as the diffusion loss, and to our introduced regularization term as "Dispersive Loss". Formally, denoting $X = \{x_i\}$ as a batch of noisy images $x_i$, the objective function of this batch is given by:

$$\mathcal{L}(X) = \mathbb{E}_{x_i \in X}[\mathcal{L}_{\text{Diff}}(x_i)] + \lambda \mathcal{L}_{\text{Disp}}(X). \tag{1}$$

Here, $\mathcal{L}_{\text{Diff}}(x_i)$ is the standard diffusion loss of one sample, and $\mathcal{L}_{\text{Disp}}(X)$ is the dispersive loss term that depends on the entire batch, with a weighting scale $\lambda$. In our practice, we do not apply any additional layer (*e.g.*, a projection head (Chen et al., 2020)), and Dispersive Loss is applied directly on the intermediate representations, requiring no extra learnable parameters.

Our method is self-contained and minimalist. In particular, it does *not* alter the implementation of the original $\mathcal{L}_{\text{Diff}}$ term: it introduces no additional sampling views, no extra data augmentation, and when $\lambda$ is zero it reduces exactly to the baseline diffusion model. This design is possible because the introduced Dispersive Loss $\mathcal{L}_{\text{Disp}}(X)$ only depends on the already-computed intermediate representations of the same input batch. This is unlike standard contrastive learning (e.g., (Chen et al., 2020)), where the additional augmentation and view may interfere with the per-sample regression objective.

Table 1: **Variants of Dispersive Loss functions**, compared against their contrastive counterparts. Each variant of Dispersive Loss can be viewed as the contrastive loss without positive pairs.

| variant | contrastive | dispersive |
|---------|-------------|------------|
| InfoNCE | $D(z_i,z_i^+)/\tau+\log\sum_j\exp(-D(z_i,z_j)/\tau)$ | $\log\mathbb{E}_{i,j}[\exp(-D(z_i,z_j)/\tau)]$ |
| Hinge | $D(z_i,z_i^+)^2+\mathbb{E}_j[\max(0,\epsilon-D(z_i,z_j))^2]$ | $\mathbb{E}_{i,j}[\max(0,\epsilon-D(z_i,z_j))^2]$ |
| Covariance | $(1-\mathrm{Cov}_{mm})^2+w\sum_{n\neq m}\mathrm{Cov}_{mn}^2$ | $\sum_{m,n}\mathrm{Cov}_{mn}^2$ |

Intuitively, Dispersive Loss behaves like a "contrastive loss *without* positive pairs". In the context of generative modeling, this formulation is plausible because the regression terms provide predefined targets for training, making the use of "positive pairs" unnecessary. This is consistent with prior research on self-supervised learning (Wang & Isola, 2020), where positive terms are interpreted as alignment objectives, and negative terms as forms of regularization. By eliminating the need for positive pairs, the loss terms can be defined on any standard batch of (independent) images.

Conceptually, Dispersive Loss can be derived from any existing contrastive loss by appropriately removing the positive terms. In this regard, the term "Dispersive Loss" does not refer to a specific implementation, but rather to a general class of objectives that encourage dispersion. We introduce several variants of Dispersive Loss functions in the following.

## 3.2 INFONCE-BASED VARIANT OF DISPERSIVE LOSS

InfoNCE (Oord et al., 2018) is a widely used and effective variant of contrastive loss in self-supervised learning (*e.g.*, (He et al., 2020; Chen et al., 2020)). As a case study, we introduce the *dispersive* counterpart of the InfoNCE loss. Formally, let $z_i = f(x_i)$ denote the intermediate representations of the generative model for an input sample $x_i$, where $f$ represents the subset of layers used to compute the intermediate representations. The original InfoNCE loss (Oord et al., 2018) can be interpreted as a categorical cross-entropy objective that encourages high similarity between positive pairs and low similarity between negative pairs:

$$\mathcal{L}_{\text{Contrast}}=-\log\frac{\exp(-\mathcal{D}(z_i,z_i^+)/\tau)}{\sum_j\exp(-\mathcal{D}(z_i,z_j)/\tau)}. \tag{2}$$

Here, $(z_i, z_i^+)$ denotes a pair of positive samples (*e.g.*, obtained by data augmentation of the same image), and $(z_i, z_j)$ denotes any pair of samples which include the positive pair and all negative pairs (*i.e.*, $i \neq j$). $\mathcal{D}$ denotes a dissimilarity function (*e.g.*, distance), and $\tau$ is a hyper-parameter known as the temperature. A commonly used form of $\mathcal{D}$ is the negative cosine similarity (Wu et al., 2018): $\mathcal{D}(z_i,z_j)=-z_i^\top z_j/(\|z_i\|\|z_j\|)$.

Inside the logarithm in Eq. (2), the numerator involves only the positive pair $(z_i, z_i^+)$, whereas the denominator includes all pairs in the batch. Following (Grill et al., 2020; Wang & Isola, 2020), we can rewrite Eq. (2) equivalently as:

$$\mathcal{L}_{\text{Contrast}}=\mathcal{D}(z_i,z_i^+)/\tau + \log\sum_j\exp(-\mathcal{D}(z_i,z_j)/\tau). \tag{3}$$

Here, the first term is similar to a *regression* objective, which minimizes the distance between $z_i$ and its target $z_i^+$. On the other hand, the second term encourages any pair of $(z_i,z_j)$ to be as distant as possible.

To construct the Dispersive Loss counterpart, we keep only the second term:

$$\mathcal{L}_{\text{Disp}} = \log\sum_j\exp(-\mathcal{D}(z_i,z_j)/\tau). \tag{4}$$

This formulation can also be viewed as a contrastive loss (Eq. (3)), where each positive pair consists of two *identical* views $z_i^+ = z_i$, making $\mathcal{D}(z_i,z_i^+)$ a constant. Eq. (4) is equivalent to:

$$\mathcal{L}_{\text{Disp}} = \log\mathbb{E}_j[\exp(-\mathcal{D}(z_i,z_j)/\tau)], \tag{5}$$

up to a constant $\log(\text{batch size})$ that does not impact optimization. Conceptually, this loss definition is based on a reference sample $z_i$. To have a form defined on a batch of samples $Z = \{z_i\}$, we follow (Wang & Isola, 2020) and redefine it as:

$$\mathcal{L}_{\text{Disp}}=\log\mathbb{E}_{i,j}[\exp(-\mathcal{D}(z_i,z_j)/\tau)]. \tag{6}$$

**Algorithm 1** Dispersive Loss (InfoNCE, $\ell_2$ dist.)

It takes as input the flattened intermediate representations $Z$ of shape $N \times D$ (with $D = H \times W \times C$).

```
def disp_loss(Z, tau):
  D = pdist(Z, p=2)**2 / Z.shape[1]
  return log(mean(exp(-D/tau)))
```

**Algorithm 2** Diffusion with Dispersive Loss

It takes as input the model's output prediction, flattened intermediate activation $Z$, denoising target, and the weight scale $\lambda$.

```
def loss(pred, Z, tgt, lamb):
  L_diff = mean((pred - tgt)**2)
  L_disp = disp_loss(Z)
  return L_diff + lamb * L_disp
```

This loss function has the same value for all samples within a batch and is computed only once per batch. In our experiments, in addition to the cosine dissimilarity, we also study the squared $\ell_2$ distance: $\mathcal{D}(z_i, z_j) = \|z_i - z_j\|_2^2$. When using this $\ell_2$ form, the Dispersive Loss can be easily computed by a few lines of code, as shown in Algorithm 1.

The InfoNCE-based Dispersive Loss as defined in Eq. (6) is similar to the *uniformity loss* in (Wang & Isola, 2020) (though we do not $\ell_2$-normalize the representations). In the context of contrastive representation learning considered in (Wang & Isola, 2020), the uniformity loss is applied to the output representation and must be paired with an alignment loss (*i.e.*, the positive terms). Our formulation goes a step further by removing the alignment term on the intermediate representations, and thereby focusing solely on the regularization perspective.

We note that we do not need to explicitly exclude the term $\mathcal{D}(z_i, z_j)$ when $j = i$. Since we do not use multiple views of the same image in one batch, this term always corresponds to a constant and minimal dissimilarity, *e.g.*, 0 in the $\ell_2$ case and –1 in the cosine case. As such, this term acts as a constant bias inside the logarithm, and its contribution becomes small when the batch size is sufficiently large. In practice, it is not necessary to exclude this term, which also simplifies the implementation.

### 3.3 OTHER VARIANTS OF DISPERSIVE LOSS

The concept of Dispersive Loss naturally extends to a broad class of contrastive loss functions beyond InfoNCE. Any objective that encourages the repulsion of negative samples may be considered a dispersive objective and instantiated as a variant of Dispersive Loss. We introduce two additional variants based on other types of contrastive loss functions. Table 1 summarizes all three variants and compares the contrastive and dispersive counterparts, as described below.

**Hinge Loss**. In the classical formulation of contrastive learning (Chopra et al., 2005), the loss function is defined as a sum of independent loss terms, each corresponding to a positive or negative pair. The loss term for a positive pair is $\mathcal{D}(z_i, z_i^+)$; the loss term for a negative pair is formulated as a squared hinge loss, *i.e.*, $\max(0, \epsilon - \mathcal{D}(z_i, z_j))^2$, where $\epsilon > 0$ is the margin. To construct the Dispersive Loss counterpart, we simply discard the loss terms for positive pairs and compute only the terms for negative pairs. See Table 1 (row 2).

**Covariance Loss**. Another class of (generalized) contrastive loss functions operates on the cross-covariance matrix of the representations. This class of loss functions encourages the cross-covariance matrix to be close to the identity matrix. As an example, we consider the loss defined in "Barlow Twins" (Zbontar et al., 2021), which calculates a cross-covariance matrix between the normalized representations of two augmented views of a batch. Denote the $D \times D$ cross-covariance as "Cov" with elements indexed by $(m, n)$. The loss encourages the diagonal elements $\text{Cov}_{mm}$ to be one, using a loss term $(1 - \text{Cov}_{mm})^2$, and off-diagonal elements $\text{Cov}_{mn}$ ($\forall m \neq n$) to be zero, using a loss term $w \sum_{m \neq n} \text{Cov}_{mn}^2$ for a weight $w$.

In our dispersive counterpart, we consider only the off-diagonal elements $\text{Cov}_{mn}$. Since we do not use augmented views, the cross-covariance reduces to the covariance matrix computed over a single-view batch. In this case, the diagonal elements $\text{Cov}_{mm}$ automatically equal one when the representations are $\ell_2$-normalized, and thus do not need to be explicitly addressed in the loss function. The resulting Dispersive Loss is simply $\sum_{m,n} \text{Cov}_{mn}^2$. See Table 1 (row 3).

### 3.4 DIFFUSION MODELS WITH DISPERSIVE LOSS

As summarized in Table 1, all variants of Dispersive Loss exhibit simpler formulations than their contrastive counterparts. More importantly, all Dispersive Loss functions are applicable to a *single-view* batch,

eliminating the need for multi-view augmentations. As a result, they can serve as plug-and-play regularizers within existing generative models, without modifying the implementation of the regression loss.

In practice, incorporating Dispersive Loss requires only *minimal* adjustments: (i) specifying the intermediate layer on which to apply the regularizer, and (ii) computing the Dispersive Loss at this layer and adding it to the original diffusion loss. Algorithm 2 presents the training pseudo-code, with a specific form of Dispersive Loss as defined in Algorithm 1. We believe that such simplicity greatly facilitates the practical adoption of our methodology, enabling its application across various generative models.

## 4 EXPERIMENTS

### 4.1 EXPERIMENT SETUP

The majority of our experiments are conducted on standard models: DiT (Peebles & Xie, 2023) and SiT (Ma et al., 2024), which respectively serve as the *de facto* diffusion-based and flow-based baselines. We faithfully follow the original implementations in (Peebles & Xie, 2023; Ma et al., 2024) and conduct experiments on the ImageNet dataset (Deng et al., 2009) at 256×256 resolution. The generative models are trained on a 32×32×4 latent space produced by a VAE tokenizer (Rombach et al., 2022). Sampling is performed using the ODE-based Heun sampler with 250 steps, following (Peebles & Xie, 2023; Ma et al., 2024). In our ablation experiments, unless specified, we train the models for 80 epochs (Ma et al., 2024) and without classifier-free guidance (CFG) (Ho & Salimans, 2022). By default, the weight scale $\lambda$ in Eq. (1) is 0.5, and the temperature $\tau$ in Eq. (6) is 0.5. The full implementation details are in Section A.

### 4.2 MAIN OBSERVATIONS ON DIFFUSION MODELS

**Dispersive *vs.* Contrastive.** In Table 2, we compare different variants of Dispersive Loss with their contrastive counterparts. To apply a contrastive loss, two views are sampled for each training example to form a positive pair. We study two strategies for adding noise to both views: (i) sampling noise *independently* for each view, following the generative model's noising policy; or (ii) sampling noise for the first view according to the same policy, and then *restricting* the second view's noise level to differ by at most 0.005. Moreover, to avoid doubling training epochs due to two-view sampling, we only apply the denoising loss to the first view, for fair comparisons with both the baseline and our method.

Table 2 shows that when using independent noise, *the contrastive loss fails to improve generation quality in all cases studied*. We hypothesize that aligning two views with substantially different noise levels impairs learning. As evidence, Table 2 shows that contrastive loss with *restricted* noise leads to improvements over the baseline in three of the four evaluated variants. These experiments suggest that contrastive learning is sensitive to the choice of data augmentation (consistent with observations in self-supervised learning (Chen et al., 2020)). *Noising, which serves as a built-in form of augmentation in diffusion models, further complicates the problem.*. While contrastive learning can be mildly *beneficial* (as shown in Table 2), the design of the additional view and the coupling of two views may limit its application.

By comparison, Table 2 shows that Dispersive Loss yields consistent improvements over the baseline, while avoiding the complications introduced by two-view sampling. Despite its simplicity, Dispersive Loss achieves *better* results than its contrastive counterpart, even when the latter uses carefully tuned noise. We note that Dispersive Loss is applied to a single-view batch, and its influence on training arises solely from its regularizing effect on the intermediate representations.

**Variants of Dispersive Loss.** The experiments in Table 2 involve four variants of Dispersive Loss (see Table 1): InfoNCE ($\ell_2$ or cosine dissimilarity), Hinge, and Covariance. As discussed, *all* variants of Dispersive Loss outperform the baseline, highlighting the generality and robustness of the approach.

Among these variants, InfoNCE with the $\ell_2$ distance performs best: it improves the FID by a substantial margin of 4.14, or relative 11.35%. This is in contrast to common practice in self-supervised learning, where cosine dissimilarity is typically preferred. We note that we do *not* apply normalization to the representations before computing this InfoNCE loss, and as a result, the distance between two samples can be arbitrarily large. We hypothesize that this design can encourage the representations to be more dispersed, leading to stronger regularization. We use the $\ell_2$-based InfoNCE in other experiments by default.

| variant | baseline | contrastive (independent noise) | contrastive (restricted noise) | dispersive |
|---|---|---|---|---|
| none | 36.49 | – | – | – |
| InfoNCE, $\ell_2$ | – | 43.66 (+19.65%) | 36.57 (+0.22%) | **32.35** (–11.35%) |
| InfoNCE, cosine | – | 41.62 (+14.06%) | 34.83 (–4.55%) | 34.33 ( –5.92%) |
| Hinge | – | 43.02 (+17.89%) | 35.14 (–3.70%) | 33.93 ( –7.02%) |
| Covariance | – | 37.85 ( +3.73%) | 35.87 (–1.70%) | 35.82 ( –1.84%) |

Table 2: **Variants of Dispersive Loss**, compared against their contrastive counterparts. We report FID-50k of SiT-B/2 on ImageNet, trained for 80 epochs, without CFG. The "baseline" entry refers to original SiT-B/2 without any contrastive or dispersive loss. The contrastive loss involves two views per training sample, with "independent" or "restricted" noise (the latter case restricts the noise level to differ by up to 0.005). Values in brackets denote the relative improvement or degradation with respect to the baseline.

| baseline | 36.49 |
|---|---|
| block 1 | 33.64 ( –7.81%) |
| block 2 | 32.98 ( –9.62%) |
| block 3 | **32.35** (–11.35%) |
| block 4 | 32.65 (–10.52%) |
| block 8 | 32.75 (–10.24%) |
| block 12 | 33.06 ( –9.93%) |
| all blocks | **32.05** (–12.17%) |

| | $\lambda = 0.25$ | $\lambda = 0.5$ | $\lambda = 1.0$ |
|---|---|---|---|
| $\tau = 2.0$ | 33.25 | 32.85 | 32.72 |
| $\tau = 1.0$ | 33.14 | **32.10** | 32.88 |
| $\tau = 0.5$ | 33.72 | 32.35 | 33.10 |
| $\tau = 0.25$ | 33.90 | 33.56 | 33.16 |

Table 3: **Block Choice for Regularization.** We apply Dispersive Loss at different blocks of the SiT-B/2 model (12-block). FID is evaluated on ImageNet. The regularizer is beneficial in all cases.

Table 4: **Loss weight $\lambda$ and temperature $\tau$.** We investigate the two hyper-parameters of Dispersive Loss with SiT-B/2 on ImageNet. All cases are substantially better than the baseline (36.49).

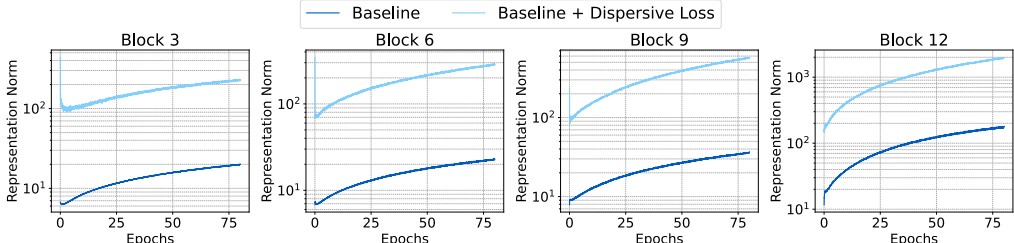

Figure 3: **Evolvement of Representation Norm.** Dispersive Loss is applied ***only at Block 3***. Dispersive Loss significantly increases the representation norm compared to the baseline—even in layers where it is not directly applied. The model is SiT-B/2 on ImageNet.

**Block Choice for Regularization.** In Table 3, we investigate the effect of Dispersive Loss at different layers (*i.e.*, Transformer blocks). Overall, *the regularizer improves over the baseline by a substantial margin in all cases studied*, showing the generality of our approach. Applying Dispersive Loss to all blocks yields the best result, while applying it to *any* single block performs nearly as well.

To take a closer look, Fig. 3 shows the $\ell_2$ norm of the representations in a model where Dispersive Loss is applied *only* at Block 3. Notably, our regularizer yields a larger representation norm at Block 3 and propagates this effect to all other blocks, even though it is not directly applied to them. This helps explain the consistent gains observed in Table 3 wherever Dispersive Loss is applied. In our other experiments, Dispersive Loss is applied to the single block at the first quarter of the total blocks.

**Loss weight $\lambda$ and temperature $\tau$.** Our method only involves two hyper-parameters that need to be specified: $\lambda$, which controls the regularization strength, and $\tau$, the temperature in InfoNCE. In Table 4, we study their effect on the results. Overall, *all* configurations studied improve upon the baseline (FID 36.49), further suggesting that the regularizer is broadly effective. Notably, we compare a wide range of temperature values $\tau$ and find that the results are surprisingly robust: this is in contrary to contrastive self-supervised learning, where performance degrades significantly when deviating from the optimal temperature (Chen et al., 2020).

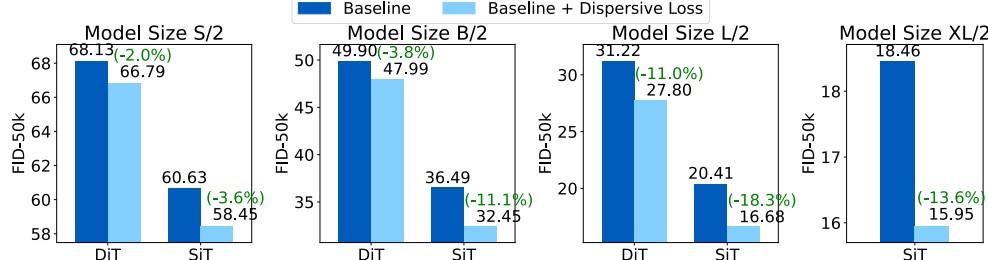

Figure 4: **Dispersive Loss for Different Models.** We evaluate DiT and SiT baselines across four model sizes, both with and without Dispersive Loss. Each subplot corresponds to a specific model size (XL is only for SiT due to limited computation). All models are trained on ImageNet for 80 epochs.

| epochs | w/o CFG | | | w/ CFG | | |
|---|---|---|---|---|---|---|
| | **baseline** | **dispersive** | $\Delta$ | **baseline** | **dispersive** | $\Delta$ |
| 80 | 18.46 | 15.95 | −2.51 (−13.6%) | 6.02 | 5.09 | −0.93 (−15.4%) |
| 140 | 14.06 | 12.08 | −1.98 (−14.0%) | 3.95 | 3.42 | −0.53 (−13.4%) |
| 200 | 12.18 | 10.64 | −1.54 (−12.6%) | 3.30 | 2.90 | −0.40 (−12.1%) |
| 400 | 10.11 | 8.81 | −1.30 (−12.8%) | 2.69 | 2.39 | −0.30 (−11.1%) |
| 800 | 8.99 | 8.08 | −0.91 (−10.1%) | 2.46 | 2.12 | −0.34 (−13.8%) |
| 800 (w/ SDE) | 8.58 | 7.71 | −0.87 (−10.1%) | 2.27 | 2.09 | −0.18 (−7.93%) |
| $\geq$1200 (w/ SDE) | 8.26 | 7.43 | −0.83 (−10.0%) | 2.06 | **1.97** | −0.09 (−4.36%) |

Table 5: **More Experiments on SiT-XL/2.** We train the largest SiT model (XL) for more epochs, with and without using CFG, to match the setup in the SiT paper (Ma et al., 2024). Results are reported as FID-50k on ImageNet 256×256. The sampler used is the ODE-based Heun method, except for the last two rows, which use the SDE-based Euler method following (Ma et al., 2024). In the last row, the baseline SiT results are taken from the original paper (Ma et al., 2024), which reports 1400-epoch training; our results with Dispersive Loss in the last row are based on 1200-epoch training. In all cases, Dispersive Loss yields substantial improvements in FID over the baseline.

| method | pre-training | additional params | external data | FID (SiT-XL/2) |
|---|---|---|---|---|
| REPA (Yu et al., 2024) | ✓ (1500 ImgNet epochs) | ✓ (1.1B params) | ✓ (142M images) | **1.80** |
| **Dispersive Loss** | ✗ | ✗ | ✗ | 1.97 |

Table 6: **System-level comparison with REPA.** While Dispersive Loss yields a smaller gain compared to its REPA counterpart, our method is *self-contained* and does not rely on any external models. In contrast, REPA uses a pre-trained DINOv2 model (Oquab et al., 2023), whose overhead is summarized in this table.

**Dispersive Loss for Different Models.** Thus far, our ablations have focused on the SiT-B/2 model. In Fig. 4, we extend our evaluation to both DiT (Peebles & Xie, 2023) and SiT (Ma et al., 2024) across four commonly used model sizes (S, B, L, XL). Once again, Dispersive Loss consistently improves performance over the baseline in all scenarios.

Interestingly, we observe that both the relative and even absolute improvements tend to be *larger* when the baseline is *stronger*. For each specific model size, both the relative and absolute improvements upon the SiT baseline are larger than those upon the DiT baseline—even though the SiT baselines are stronger than their DiT counterparts. Similarly, L-size models exhibit greater relative improvements compared to B- or S-size models (this trend plateaus at the XL size, likely because there is less room for further improvement). Overall, this trend provides strong evidence that the primary effect of Dispersive Loss lies in *regularization*. As larger and more capable models are more prone to overfitting, they tend to benefit more from effective regularization.

**More Experiments on SiT-XL/2.** In Table 5, we train SiT-XL/2 while faithfully following all practices described in the original SiT paper (Ma et al., 2024). Specifically, we train the model for more epochs, both with and without classifier-free guidance (CFG) (Ho & Salimans, 2022). Both ODE-based and

Figure 5: **Qualitative results.** We present curated samples generated from SiT-XL/2 with Dispersive Loss.

| model | epochs | **baseline** | **dispersive** | $\Delta$ |
|---|---|---|---|---|
| MF-B/4 | 80 | 18.78 | 17.61 | −6.23% |
| MF-B/2 | 80 | 9.77 | 8.97 | −8.18% |
| MF-B/2 | 240 | 6.17 | 5.69 | −7.77% |
| MF-XL/2 | 240 | 3.43 | **3.21** | −6.41% |

| method | params | step | NFE | FID |
|---|---|---|---|---|
| iCT-XL/2 | 675M | 1 | 1 | 34.24 |
| Shortcut-XL/2 | 675M | 1 | 1 | 10.60 |
| IMM-XL/2 | 675M | 1 | 2 | 7.77 |
| MeanFlow-XL/2 | 676M | 1 | 1 | 3.43 |
| MeanFlow-XL/2 + **Disp** | 676M | 1 | 1 | **3.21** |

Table 7: **Dispersive Loss for One-Step Generation**. All results are FID-50k on ImageNet 256×256. **Left**: Dispersive Loss consistently improves the MeanFlow (Geng et al., 2025) baseline. **Right**: Comparison of state-of-the-art one-step diffusion/flow-based models.

SDE-based samplers are investigated. Overall, our method proves beneficial across all settings, even as the baselines become stronger. Some example images generated by this model are in Fig. 5.

**System-level Comparison with REPA**. Our method is related to REPA (Yu et al., 2024). While our regularizer operates directly on the model's internal representations, REPA aligns them with those from an external model. As such, for a fair comparison, both the additional computational overhead and the external sources of information should be taken into account, summarized in Table 6. REPA relies on a pre-trained DINOv2 model (Oquab et al., 2023), which itself is distilled from a 1.1B-parameter backbone trained on 142 million curated images for the equivalent of 1500 ImageNet epochs. In comparison, our method is entirely self-contained, requiring no pre-training, no external data, and no additional model parameters. Our method is readily applicable when scaling up training to larger models and datasets, and we expect that the regularization effect will be desirable in such regimes.

### 4.3 ONE-STEP GENERATION MODELS

Our method can be directly generalized to *one-step* diffusion-based generative models. In Table 7 (left), we apply Dispersive Loss to the recent MeanFlow model (Geng et al., 2025) (implementation details in Section A), and observe consistent improvements. Table 7 (right) compares these results with the latest one-step diffusion/flow-based models (Song & Dhariwal; Frans et al.; Zhou et al., 2025; Geng et al., 2025), showing that our method enhances MeanFlow and achieves a new state of the art.

## 5 CONCLUSION

In this work, we have proposed Dispersive Loss, an objective that regularizes the internal representations of diffusion models. Without relying on additional pre-training, extra parameters, or external data, our approach demonstrates that representation regularization can effectively enhance generative modeling.

A key principle guiding our design is to introduce minimal or no interference with the sampling process of the original training objective. This allows us to fully preserve the original diffusion training strategy, which can be critical for maintaining generative performance. Nevertheless, a standalone, plug-and-play regularizer is not only desirable for generative modeling; its favorable properties may also benefit other applications, including image recognition. We hope our regularizer will generalize to broader scenarios, a direction we leave for future exploration.

## ETHICS AND REPRODUCIBILITY STATEMENTS

We strictly adhere to the ICLR Code of Ethics. We use open-sourced datasets and models for all of our experiments, and we strictly follow existing evaluation protocols.

For reproducibility, we include our code in the supplementary materials. We also include a detailed README file that describes how to reproduce our results.

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

| model | S/2 | B/2 | L/2 | XL/2 |
|---|---|---|---|---|
| **model configurations** | | | | |
| params (M) | 33 | 130 | 458 | 675 |
| depth | 12 | 12 | 24 | 28 |
| hidden dim | 384 | 768 | 1024 | 1152 |
| patch size | 2 | 2 | 2 | 2 |
| heads | 6 | 12 | 16 | 16 |
| **training configurations** | | | | |
| epochs | 80 | 80 | 80 | 80 - 1200 |
| batch size | | | 256 | |
| optimizer | | | AdamW | |
| optimizer $\beta_1$ | | | 0.9 | |
| optimizer $\beta_2$ | | | 0.95 | |
| weight decay | | | 0.0 | |
| learning rate (lr) | | | $1 \times 10^{-4}$ | |
| lr schedule | | | constant | |
| lr warmup | | | none | |
| **ODE sampling** | | | | |
| steps | | | 250 | |
| sampler | | | Heun | |
| $t$ schedule | | | linear | |
| last step size | | | N/A | |
| **SDE sampling** | | | | |
| steps | | | 250 | |
| sampler | | | Euler | |
| $t$ schedule | | | linear | |
| last step size | | | 0.04 | |
| **Dispersive Loss** | | | | |
| regularization loss weight $\lambda$ | | | 0.5 | |
| temperature $\tau$ | | | 0.5 | |

Table 8: **SiT and DiT Configurations on ImageNet.**

## A  IMPLEMENTATION

**SiT and DiT Experiments.** We faithfully follow the SiT/DiT codebase for ImageNet experiments. We employ the AdamW optimizer with a constant learning rate of $1 \times 10^{-4}$, $(\beta_1, \beta_2) = (0.9, 0.95)$, and no weight decay. For ODE sampling, we use the Heun solver with 250 steps, following the SiT paper Ma et al. (2024); for SDE sampling, we apply an Euler–Maruyama solver with the default drift and diffusion schedule from SiT Ma et al. (2024). We re-implement both SiT and DiT in JAX. All experiments are run on 32 TPU-v3/v4 cores. The configuration details are in Table 8.

**MeanFlow Experiments.** Our MeanFlow experiments use the exact same codebase shared by the authors of Geng et al. (2025). In Table 7, we evaluate three models: B/4, B/2, and XL/2, with and without Dispersive Loss. The B/4 and B/2 models are all trained from scratch. For the XL/2 model, due to limited computation resources, we take the MeanFlow-XL/2 checkpoint at 180 epochs and then apply Dispersive Loss in the remaining 60 epochs; we expect training this model from scratch will lead to better results.

For the MeanFlow baselines, all configurations of MF-B/2 and MF-XL/2 follow those in Geng et al. (2025). For MF-B/4, we adopt the setting of $\omega' = 2.0$ and $\kappa = 0.5$ using the notation of Geng et al. (2025). Dispersive Loss is applied on the same single block as in our SiT experiments. We set the regularization weight $\lambda$ as 0.25 for B/4, 1.0 for B/2, and 1.5 for XL/2.

# B  ADDITIONAL EXPERIMENTS

## B.1  INCEPTION SCORES

| model | epochs | w/o CFG | | | w/ CFG | | |
|---|---|---|---|---|---|---|---|
| | | **baseline** | **dispersive** | $\Delta$ | **baseline** | **dispersive** | $\Delta$ |
| B/2 | 80 | 42.79 | 48.79 | +14.02% | 93.26 | 108.23 | +16.05% |
| XL/2 | 800 | 141.28 | 147.35 | +4.11% | 270.44 | 281.26 | +4.00% |

Table 9: **Inception Scores on ImageNet**.

In Table 9 we report the Inception Scores of SiT models on ImageNet. Similar to observations on FID (Table 5), Dispersive Loss improves Inception Scores (higher being better) over the baseline SiT models.

## B.2  CIFAR-10 EXPERIMENTS

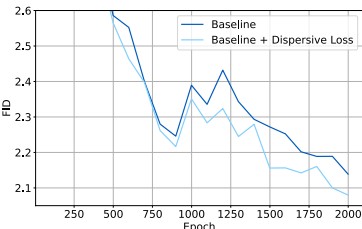

Figure 6: **CIFAR-10 Result.**

We also evaluate our approach on non-transformer architectures in the CIFAR-10 dataset Krizhevsky (2009), where the commonly used network architecture is Unet Ronneberger et al. (2015). The experiments are based on the publicly available code of Flow Matching Lipman et al. (2024).[2]  We use the same hyper-parameters as the original codebase, and our rerunning of the Flow Matching baseline has 2.13 FID. As a reference, the original repo reported 2.07 FID.

Figure 6 represents the evolvement of the FID during training, comparing the baseline against the one with Dispersive Loss, using the same random seed. The Dispersive Loss is applied at Residual Block 15 of the Unet. It can be observed that our regularizer yields consistent gains over the baseline throughout training. Our final result is 2.07 FID, which is better than the 2.13 FID of our reproduced baseline. While the absolute numbers may depend on subtle variations in the runtime environment, Figure 6 shows that the gains are persistent when evaluated under a controlled environment.

# C  VISUALIZATIONS

## C.1  QUALITATIVE COMPARISON

To qualitatively assess the effect of Dispersive Loss on generated samples, we present a visual comparison of samples generated with and without Dispersive Loss. We use the SiT-XL/2 model to generate these samples. As shown in Fig. 7, Dispersive Loss leads to better lighting and geometry in generated samples.

## C.2  TRAINING DYNAMICS

For more coherent understanding of the effect of Dispersive Loss, we present the evolution of training loss in Fig. 8. We report the first 50000 steps of training on SiT-B/4 model. We find that the value of Dispersive Loss quickly drops during the initial stage of training and maintains a constant value for the

[2]https://github.com/facebookresearch/flow_matching

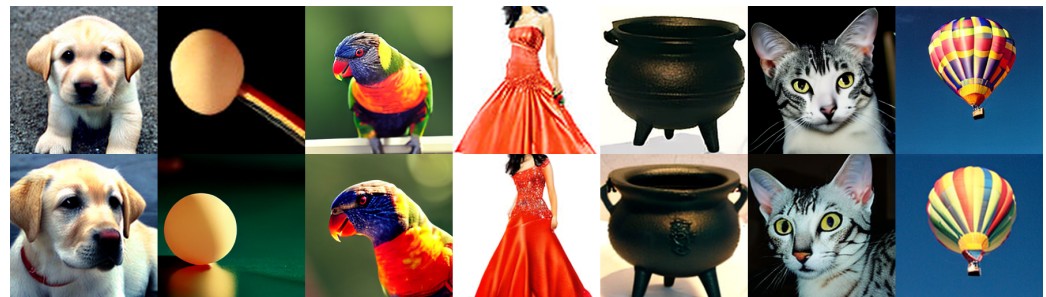

Figure 7: **Visual Comparison With Dispersive Loss.** We present samples generated with the exact set of initial noises, under models trained without (top) and with (bottom) Dispersive Loss. Dispersive Loss leads to more coherent geometric structure and more reasonable lighting and reflections.

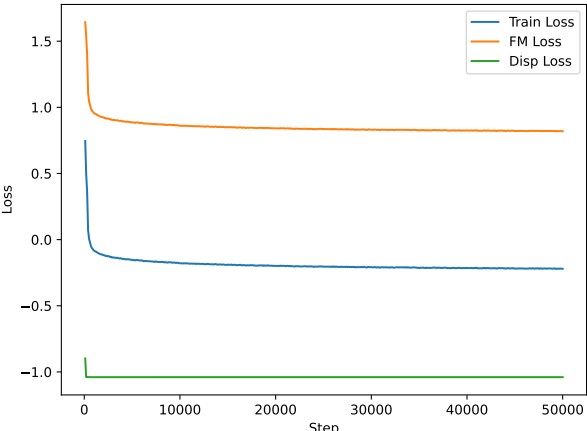

Figure 8: **Loss Curves.** We present the Flow Matching Loss curve, the Dispersive Loss curve, and the summed total training loss curve during first 50000 steps of training. The value of Dispersive Loss quickly drops at first then stays constant, which fits our expectation of the behavior of a regularizer.

remainder of the training. This fits our expectation of Dispersive Loss as a regularizer that does not significantly interfere training.

## D  LLM USAGE

LLMs are only used to correct grammar mistakes and are not used for research ideation or writing.

