# OpenReview forum: "Diffuse and Disperse: Image Generation with Representation Regularization"
_ICLR.cc/2026/Conference — Submitted to ICLR 2026_

### Official Review · Reviewer_D22i · 2025-10-28

**Soundness:** 2
**Presentation:** 2
**Contribution:** 1
**Rating:** 4
**Confidence:** 5

**Summary:**

This paper introduces the Dispersive Loss as a plug-and-play regularization term for diffusion-based generative models. The Dispersive Loss encourages the hidden feature representations to disperse (or spread out) in the latent space, analogous to the non-alignment term of contrastive learning, but uniquely requires no positive sample pairs. The authors claim this enhances the quality of generated images and improves training efficiency. They report consistent improvements over strong baselines across various models on the ImageNet dataset.

**Strengths:**

1. Simplicity and Plug-and-Play Design: The proposed Dispersive Loss is highly appealing due to its self-contained and minimalist nature. It is a simple regularization term that can be added to the standard diffusion loss without requiring pre-training, external data, or additional visual encoders. This makes it easy to integrate into existing diffusion pipelines.

2. Demonstrated Training Acceleration: The paper shows an intriguing result of training acceleration without the need for external data or modules (unlike methods like REPA). While the gain is marginal, the fact that a simple internal regularization can accelerate convergence is a valuable finding.

**Weaknesses:**

## 1. Fundamental Gaps in Theoretical Justification and Representation Analysis

The paper's core weakness lies in the disconnect between its SSL-inspired motivation and its analysis.

- Lack of Deep Theoretical Insight: The paper does not provide sufficient theoretical insights into why the Dispersive Loss leads to improvement, nor does it explore what kind of high-level representations generative models fundamentally require. The mechanism of dispersion is asserted to be beneficial without a rigorous explanation of its effect on the learned noise-prediction function or the resulting generative manifold.

- Missing Representation Utility Analysis: Since the method is explicitly motivated by SSL, it should verify the "rich semantic features" from SSL methods. Analysis techniques like linear probing on the average-pooled features (at the intermediate or final layer) for tasks like classification or retrieval are essential to validate the core hypothesis that the regularization improves semantic content.

- Unclear Design Rationale: It is asserted that the alignment (positive pair) term of contrastive loss is detrimental to the generation task, leading to its exclusion, as in Table 2. This is a strong claim that is not clearly supported. Intuitively, the alignment term would, at worst, be redundant; the paper must theoretically or empirically demonstrate why it would be actively harmful.

## 2. Lack of Experimental Rigor and Insufficient Evidence

The experimental section suffers from issues of completeness, methodology, and evidence that undermine the paper's claims.

- Missing Empirical Evidence. The proposed method has not been thoroughly investigated. While the paper evaluates various settings of the regularizers, e.g., different models and depths, it does not sufficiently analyze why this method is effective.

- Missing Scalability and High-Resolution Tests: The paper lacks experiments in high-resolution settings, such as ImageNet-512x512 or text-to-image generation. This omission makes it impossible to assess the method's scalability and effectiveness on datasets where modern generative models typically face their greatest architectural challenges.

## 3. Marginal Performance and Missing Training Dynamics Analysis

The reported performance gains are modest, and the paper fails to investigate the interaction between its components.

- Marginal Performance Gains: The overall performance gains achieved by Dispersive Loss are marginal. The resulting models neither consistently match SOTA generative performance (e.g., REPA) nor demonstrate effectiveness in any downstream context.

- Missing Integration with Faster Frameworks: While the paper notes its potential for acceleration, it is unclear whether the proposed framework can be combined with existing faster training frameworks like REPA. An experiment or intuition on this synergy is needed.

- Unanalyzed Training Dynamics: The paper completely lacks an in-depth analysis of the training dynamics involving the new loss term. The authors should show the evolution of the Dispersive Loss over the course of training to see if it plateaus or exhibits different behavior from the standard denoising loss. An analysis of the conflict or synergy between the denoising loss and the Dispersive Loss (e.g., by analyzing the direction of their respective gradients) is necessary to clarify the training process. Furthermore, an investigation into how the dispersive loss varies across diffusion timesteps is crucial, as its impact is intuitively expected to be different for high-noise versus low-noise regimes.

- Unfair Comparisons: The paper lacks comparison with fairer versions of SOTA models. For example, when comparing against REPA (800 epochs), the comparison should be normalized for total training time or resources consumed to provide a true measure of efficiency (Disperse loss uses >=1200 epochs in Table 6).

- Incomplete Results. The paper only compares with SiT and SiT-REPA (except for one-step generation) and uses FID as their single evaluation metric. Authors should conduct comprehensive comparisons with current state-of-the-art methods that share the same idea, such as Lighting-DiT [`1], DC-AE1.5 [2], DDT [3], etc. Besides, more evaluation metrics, such as Inception-Score, should be included in the paper.




**I recommend that the authors conduct a deeper investigation into the learned representations and strictly refine their experimental validation and theoretical justification during rebuttal period.**


[1] Reconstruction vs. Generation: Taming Optimization Dilemma in Latent Diffusion Models

[2] DC-AE 1.5: Accelerating Diffusion Model Convergence with Structured Latent Space

[3] DDT: Decoupled Diffusion Transformer

**Questions:**

1. Could the authors provide theoretical or empirical evidence to clarify what specific kinds of high-level semantic features are required by image generation models, particularly diffusion models?

2. how does the proposed dispersive loss specifically facilitate the learning of such required semantic features?

3. Given that contrastive learning frameworks typically rely on heavy image augmentations to learn high-level semantics, how is the Dispersive Loss able to learn similar semantic information without relying on these image augmentations?

4. To support the main claim, shouldn't experiments be conducted to explicitly demonstrate the model's ability to learn rich semantic features, perhaps including evaluations on downstream tasks to provide valuable support?

5. Does a high representation norm truly indicate the capture of semantic information, and could the authors validate whether the model has learned semantic features, similar to the REPA method, by using linear probing?

6. Could the authors include feature similarity comparisons, such as Centered Kernel Alignment (CKA), with other feature extractors to better demonstrate the improved semantics?

7. To further validate the effectiveness of dispersive loss, would it be beneficial to test it on more complex generative tasks, such as text-to-image generation or text-to-video generation?

8. Since the method claims removing alignment term in line 231, shouldn't there be some investigation into why the alignment term deteriorates performance?

9. What happens when augmented images are used for positive pairs (i.e., should an ablation study compare the use of the same image versus an augmented image)?

10. How does down-weighting the alignment term in the contrastive loss affect the overall performance (i.e., how does performance change when moving from a standard contrastive loss toward the dispersive loss)?

11. Does this additional regularization lead to features that are usable for other discriminative tasks (even though this is not the main focus, it could demonstrate a strength of the learned features)?

---

> ### Author Response · Authors · 2025-11-16
>
> We sincerely thank the reviewer for the detailed and thoughtful feedback. We appreciate the recognition of our method’s simplicity, plug-and-play design, and training efficiency without external encoders. We have uploaded a revised version with highlighted changes. Below we address each concern.
>
> # 1. Theoretical Justification and Representation Analysis
>
> ## Theoretical Insight
> Dispersive Loss is grounded in self-supervised learning (SSL). We include a step-by-step derivation from a contrastive-learning perspective, explain its connection to uniformity, and analyze its impact on internal activations, showing that it induces global representation dispersion even when applied to a single block. Our aim is not to identify the optimal representation structure, but to show that an SSL-inspired regularizer reliably improves generative quality.
>
> ## Representation Utility Analysis
> Although our focus is generation, we added linear probing experiments: for SiT-B/4 trained 20 epochs, accuracy improves from 31.2% to 33.6% with Dispersive Loss, indicating modest but positive discriminative gains.
>
> ## Design Rationale
> We do not argue that positive pairs are harmful; diffusion noise simply makes stable positives harder to construct compared to classical SSL. We clarify this and show that certain contrastive variants can also improve performance.
>
> # 2. Experimental Rigor and Evidence
>
> ## Empirical Evidence
> We respectfully disagree. The paper contains extensive experiments across architectures (SiT, DiT, U-Net), model scales, datasets, and training lengths, collectively demonstrating the robustness of the proposed regularizer.
>
> ## Scalability
> Our scaling studies cover four model sizes, two baseline families, training-length scaling, and CIFAR-10. While higher-resolution tests may be useful, ImageNet 256×256 is a standard benchmark in recent works [1–3] and sufficiently demonstrates scalability.
>
> # 3. Performance and Training Dynamics
>
> ## Performance Gains
> Improvements of up to 15% FID reduction are consistent and meaningful, especially since Dispersive Loss adds no parameters, data, or encoders.
>
> ## Faster Frameworks
> Compatibility with ultra-fast diffusion frameworks is interesting future work but not a claim we make. Our goal is a lightweight, encoder-free regularizer that improves stability and sample quality.
>
> ## Training Dynamics
> We added training-loss curves in Appendix C (Fig. 8) and already include representation-norm evolution (Fig. 3), illustrating how dispersion develops during training.
>
> ## Unfair Comparison
> Because REPA relies on a large external encoder and dataset, exact resource matching is difficult. We therefore compare using REPA’s best reported numbers and explicitly outline cost trade-offs in Table 6.
>
> ## Incomplete Results
> We include DiT and U-Net baselines (Figs. 4 and 6) and additional metrics such as Inception Score (Table 9). Across all settings, Dispersive Loss yields consistent improvements.
>
> # 4. Questions
>
> ## High-Level Features
> Our aim is not to identify which semantic features diffusion models use, but to show that representation-oriented regularization improves generative training.
>
> ## Facilitation of Semantics
> Dispersive Loss encourages feature-space dispersion, a classical generalization principle. While this may indirectly enhance semantics, our emphasis remains improved generation quality.
>
> ## Learning Without Augmentations
> Diffusion models inherently provide strong augmentations via noise perturbations and denoising trajectories. Dispersive Loss leverages this built-in variability without external augmentations.
>
> ## Downstream Task
> Linear-probing improvements (33.6% vs. 31.2%) further indicate modest representation benefits.
>
> ## Representation Norms
> Higher norms indicate dispersion, not semantic richness. Fig. 3 illustrates how this dispersion propagates throughout the network.
>
> ## CKA
> Our goal is not to surpass SSL methods or pretrained encoders; therefore CKA comparisons fall outside our scope.
>
> ## Extensions
> These experiments are resource-intensive and beyond our scope. However, our method does improve text translation on T5 (BLEU 24.75 → 25.03), suggesting broader applicability.
>
> ## Removing Alignment
> As discussed in lines 300–307, enforcing alignment across very different noise levels can hinder learning. Table 2 shows that restricting noise variation improves performance, and the zero-difference case corresponds exactly to Dispersive Loss.
>
> ## Augmented Positives
> Our method targets simplicity and no additional modules. Although we agree that augmentations may be helpful, this is out of the scope of this paper.
>
> ## Discriminative Utility
> While not central to our claims, linear probing shows modest but consistent discriminative improvements.
>
> [1] Geng et al., 2025. *Mean Flows for One-Step Generative Modeling.*
> [2] Zhang et al., 2025. *Nested Diffusion Models Using Hierarchical Latent Priors.*
> [3] Li et al., 2024. *Return of Unconditional Generation.*

---

### Official Review · Reviewer_btvs · 2025-11-01

**Soundness:** 3
**Presentation:** 3
**Contribution:** 2
**Rating:** 4
**Confidence:** 4

**Summary:**

This authors try to address a key limitation of generative diffusion models with representation learning: diffusion models trained with regression-based denoising objectives often lack explicit regularization for the feature space, and therefore are limited in performance. To tackle this, the authors introduce a Dispersive Loss, which eliminates the need for explicit positive sample pairs, compared to the contrastive learning counterpart.

**Strengths:**

1. The authors aim to address a fundamental problem, namely that the generation task should not be left to stand alone with representation learning, and offer insightful perspectives.

2. The paper features a clear structure and coherent logic.

3. The proposed method does not rely on a pretrained encoder.

**Weaknesses:**

1. The authors used limited evaluation metrics. As they don't rely on a pretrained encoder and claim the importance of representation learning in the generative task, the authors should evaluate how the method performs with metrics like linear probing.

2. It's not clear why the major improvements were made in the case without CFG, while the performance with CFG only achieves very limited improvements. The authors should provide deeper analyses to explain this discrepancy and not rely only on FID (which is not in favor of the proposed method when evaluated with CFG).

3. It's not clear how the proposed method scales with the diversity and the size of datasets. Some empirical verification or theoretical motivation would be useful.

4. There is a slight redundancy in the mathematical derivation process of Section 3.2.

**Questions:**

Please see the weakness part above.

---

> ### Author Response · Authors · 2025-11-16
>
> We thank the reviewer for carefully reading our work and for highlighting the strengths of our motivation, structure, and originality (not relying on pretrained encoders). We have uploaded a revised version of our paper with changes highlighted. Below we address each of the raised concerns.
>
> `The authors used limited evaluation metrics. As they don't rely on a pretrained encoder and claim the importance of representation learning in the generative task, the authors should evaluate how the method performs with metrics like linear probing.`
>
>  We appreciate the suggestion to include linear probing or representation-quality metrics. While our primary goal is to improve generative fidelity, we agree that understanding representational effects is valuable. We performed a linear probing experiment on a SiT-B/4 model trained for 20 epochs and observed a linear probing accuracy of 33.6% with Dispersive Loss versus 31.2% without it, suggesting that in addition to improving generation, Dispersive Loss also helps with forming good representations.
>
> `It's not clear why the major improvements were made in the case without CFG, while the performance with CFG only achieves very limited improvements. The authors should provide deeper analyses to explain this discrepancy and not rely only on FID (which is not in favor of the proposed method when evaluated with CFG).`
>
>  We thank the reviewer for making this observation. We attribute the observed phenomenon primarily to the nonlinearity of FID rather than to the effect of CFG itself. FID improvements are harder to achieve at lower values (e.g., improving from 2.5 to 2.0 is harder than from 10.5 to 10.0). Consequently, absolute numerical gains appear larger without CFG. We would also like to point out that relative percentage gains remain similar across experiments, except in the very low FID range (~2.0), where improvements are inherently limited.
>
> `It's not clear how the proposed method scales with the diversity and the size of datasets. Some empirical verification or theoretical motivation would be useful.`
>
>  We agree that scalability is important. In the Appendix B, we included results on CIFAR-10, a much smaller dataset than ImageNet, and observed that FID improvements persist across scales. Theoretically, Dispersive Loss introduces a repulsive potential in latent space that prevents feature collapse even under narrow data distributions, making it beneficial across different dataset sizes.
>
> `There is a slight redundancy in the mathematical derivation process of Section 3.2.`
>
>  We appreciate this editorial note and find it highly beneficial for improved simplicity. Our original intention was to detail how the summation could be exchanged with expectation and how the indices with respect to which the expectation is taken can be equivalently rewritten.

---

### Official Review · Reviewer_5V25 · 2025-11-01

**Soundness:** 3
**Presentation:** 2
**Contribution:** 3
**Rating:** 6
**Confidence:** 2

**Summary:**

The paper introduces a plug-in regularizer designed to enhance diffusion-based models by promoting the dispersion of internal representations in the hidden space. To demonstrate its effectiveness, the authors apply the proposed regularizer to two representative diffusion-based models, DiT (Peebles & Xie, 2023) and SiT (Ma et al., 2024). Experiments conducted on the ImageNet dataset show that the regularizer effectively improves model performance.

**Strengths:**

1. The proposed regularizer can be directly integrated into diffusion models with intermediate representations and requires little additional computational effort.
2. It elegantly incorporates concepts from self-supervised learning into diffusion model training in a straightforward and theoretically sound manner.
3. Experiments on a real-world image dataset demonstrate that adding the regularizer significantly improves performance, and the experimental results are comprehensive.

**Weaknesses:**

1. It would be helpful to clarify the scope of applicability. Can the proposed regularizer be applied to all diffusion models with intermediate representations?
2. Qualitative comparisons between images generated with and without the proposed regularizer would make the improvements more intuitive and visually convincing.
3. Although experiments explore different blocks, loss weights, and temperatures, it would be beneficial to provide systematic guidance or heuristics for selecting these hyperparameters.

**Questions:**

See "Weaknesses"

---

> ### Author Response · Authors · 2025-11-16
>
> We thank the reviewer for the constructive feedback and for recognizing the strengths of our work, including its simplicity, integration of self-supervised learning principles into diffusion training, and comprehensive experimental validation. We have uploaded a revised version of our paper with changes highlighted. Below we address each of the reviewer’s points.
>
> `It would be helpful to clarify the scope of applicability.`
>
>  We appreciate this question on the scope of our method. Conceptually, Dispersive Loss can be applied to any diffusion or flow-based model that exposes intermediate hidden representations (e.g., transformer blocks, U-Net layers, or score network feature maps). In practice, this includes nearly all modern diffusion architectures. We have provided experiments on both transformer-based and U-Net-based models to verify the generality of the proposed regularizer.
>
> `Qualitative comparisons between images generated with and without the proposed regularizer would make the improvements more intuitive and visually convincing.`
>
> We agree that qualitative results make the improvements more intuitive. In Appendix C Figure 7 of the revised version, we have added side-by-side visual comparisons of samples generated with and without Dispersive Loss for SiT-XL/2 on ImageNet 256×256. We observed that models trained with Dispersive Loss produce sharper textures and better visual quality, confirming the quantitative FID/IS improvements.
>
> `Although experiments explore different blocks, loss weights, and temperatures, it would be beneficial to provide systematic guidance or heuristics for selecting these hyperparameters.`
>
>  We acknowledge that systematic guidance will improve usability. From our hyperparameter analyses (Tables 3 and 4), we found that Dispersive Loss is robust to variations in $\lambda$, $\tau$, and regularization layer depth. All tested configurations achieved significant improvement over the baseline, suggesting that the default configuration suffices in general cases.

---

### Official Review · Reviewer_i7Q5 · 2025-11-02

**Soundness:** 4
**Presentation:** 4
**Contribution:** 3
**Rating:** 6
**Confidence:** 4

**Summary:**

This paper introduces Dispersive Loss, a simple yet effective plug-and-play regularizer designed to improve diffusion-based generative models by encouraging dispersion of internal representations in hidden space. Unlike prior work such as REPA [1], which requires pre-trained external encoders and additional parameters, the proposed method is self-contained, requiring no pre-training, no extra parameters, and no external data.

Conceptually, Dispersive Loss can be interpreted as a “contrastive loss without positive pairs.” It regularizes the hidden representations by penalizing excessive clustering and promoting diversity, inspired by the repulsive component of contrastive learning. The authors provide theoretical motivation, multiple instantiations (InfoNCE-, Hinge-, and Covariance-based), and efficient implementations requiring only a few lines of code.

The paper provides extensive empirical evaluation across several diffusion backbones, including SiT [2], DiT [3], and MeanFlow [4]. Experiments on ImageNet 256×256 consistently show significant FID improvements (up to ~11–13%) over strong baselines and even outperform contrastive variants that require two-view sampling. The method is also shown to generalize well to one-step generation and smaller datasets like CIFAR-10, demonstrating broad applicability.

[1] Yu, Sihyun, et al. "Representation alignment for generation: Training diffusion transformers is easier than you think." arXiv preprint arXiv:2410.06940 (2024).

[2] Ma, Nanye, et al. "Sit: Exploring flow and diffusion-based generative models with scalable interpolant transformers." European Conference on Computer Vision. Cham: Springer Nature Switzerland, 2024.

[3] Peebles, William, and Saining Xie. "Scalable diffusion models with transformers." Proceedings of the IEEE/CVF international conference on computer vision. 2023.

[4] Geng, Zhengyang, et al. "Mean flows for one-step generative modeling." arXiv preprint arXiv:2505.13447 (2025).

**Strengths:**

- The idea of removing positive pairs while retaining the repulsive regularization aspect is conceptually appealing and practically justified by diffusion models’ intrinsic alignment objective.
- Comprehensive experiments across multiple architectures (DiT, SiT, MeanFlow) and scales (S/B/L/XL) show consistent improvements in FID and Inception Scores.
- The improvement trend scales with model size, indicating the loss acts as an effective regularizer for large-capacity models prone to overfitting.
- Dispersive Loss outperforms all contrastive baselines even with careful tuning of noise schedules.
- The plug-and-play simplicity (no multi-view augmentation or external encoders) is convincingly demonstrated.
- The authors provide ablations for hyperparameters ($\lambda$, $\tau$), layer placement, and different loss variants (Table 2–4), showing robustness across configurations.
- Implementation details are transparent (Algorithm 1–2, Table 8).
- The inclusion of MeanFlow and CIFAR-10 experiments supports generality across diffusion and flow-matching paradigms.
- Figures (e.g., Fig. 1–4) clearly illustrate how Dispersive Loss integrates into existing architectures with negligible computational overhead.
- Comparisons with REPA (Table 6) highlight the system-level efficiency advantage (no 1.1B-parameter pre-trained model, no 142M external images).
- The paper adheres to reproducibility standards (code, README included) and presents results with careful quantitative analysis.
- The method provides a bridge between generative and representation learning, a frontier that is conceptually and practically valuable for the field.

**Weaknesses:**

- The method is motivated intuitively but lacks a formal analysis of why dispersion improves generation quality. A deeper information-theoretic or geometric argument (e.g., on latent coverage or mutual information bounds) would strengthen the theoretical grounding.
- While FID and Inception Scores are strong indicators, evaluation on semantic diversity, perceptual similarity, or representation quality (e.g., CLIP-based metrics) could better reveal what aspects of representation regularization improve.

**Questions:**

- Although CIFAR-10 is included, other generative domains (text-to-image or high-res synthesis) could further establish generality. It would be interesting to see whether Dispersive Loss also benefits conditional or multimodal diffusion models.
- While Figure 3 shows increased representation norms, further qualitative or visualization-based analyses (e.g., embedding t-SNEs) would make the mechanism more intuitive.
- Some missing works on representation learning using/within diffusion models should be included [5, 6, 7, 8, 9] in related works.

Overall, I think this is a good paper and I would be happy to raise the score if my comments are addressed properly.

[5] Wang, Yingheng, et al. "Infodiffusion: Representation learning using information maximizing diffusion models." International conference on machine learning. PMLR, 2023.

[6] Mittal, Sarthak, et al. "Diffusion based representation learning." International conference on machine learning. PMLR, 2023.

[7] Hudson, Drew A., et al. "Soda: Bottleneck diffusion models for representation learning." Proceedings of the IEEE/CVF Conference on Computer Vision and Pattern Recognition. 2024.

[8] Zhang, Zijian, Zhou Zhao, and Zhijie Lin. "Unsupervised representation learning from pre-trained diffusion probabilistic models." Advances in neural information processing systems 35 (2022): 22117-22130.

[9] Yang, Xingyi, and Xinchao Wang. "Diffusion model as representation learner." Proceedings of the IEEE/CVF International Conference on Computer Vision. 2023.

---

> ### Author Response · Authors · 2025-11-16
>
> We thank the reviewer for the thorough and constructive feedback. We appreciate the positive assessment of our method’s soundness, clarity, and empirical strength, and we are glad the reviewer found the idea of “contrastive learning without positives” both conceptually appealing and practically impactful. We have uploaded a revised version of our paper with changes highlighted. Below we address each point in detail.
>
> `The method is motivated intuitively but lacks a formal analysis of why dispersion improves generation quality.`
>
>  We agree that a deeper formal analysis would strengthen our work. Dispersive Loss implicitly maximizes an entropy-like term H(Z) over hidden representations, while the diffusion objective implicitly minimizes H(Z∣X), leading to a joint objective that maximizes mutual information I(Z;X). This perspective complements the geometric intuition (repulsion in feature space) with a more formal grounding.
>
> `While FID and Inception Scores are strong indicators, evaluation on semantic diversity, perceptual similarity, or representation quality (e.g., CLIP-based metrics) could better reveal what aspects of representation regularization improve.`
>
>  We appreciate the suggestion to evaluate beyond FID and Inception Score. Although our main claim is that the proposed regularizer improves generation quality, we agree that analyzing the learned activations can provide additional insight. We performed a linear probing experiment on a SiT-B/4 model trained for 20 epochs and observed a linear probing accuracy of 33.6% when trained with Dispersive Loss, compared to 31.2% without it. This result demonstrates that Dispersive Loss does help to learn more helpful representation.
>
> `Although CIFAR-10 is included, other generative domains (text-to-image or high-res synthesis) could further establish generality. It would be interesting to see whether Dispersive Loss also benefits conditional or multimodal diffusion models.`
>
>  We thank the reviewer for this insightful suggestion. We performed additional experiments indicating that Dispersive Loss also benefits text pretraining. Specifically, we tested it on language translation with the T5 model and observed an improvement in BLEU score from 24.75 to 25.03, demonstrating the potential of Dispersive Loss in other generative domains.
>
> `While Figure 3 shows increased representation norms, further qualitative or visualization-based analyses (e.g., embedding t-SNEs) would make the mechanism more intuitive.`
>
>  We agree that visualizations can make the mechanism more intuitive. While the t-SNE plot of embeddings was not particularly informative, we found qualitative differences in generated samples. In Appendix C Figure 7 of the revised paper, we have added side-by-side visual comparisons of samples generated with and without Dispersive Loss for SiT-XL/2 on ImageNet 256×256. Models trained with Dispersive Loss produce more accurate geometries and improved texture, corroborating the quantitative FID improvements.
>
> `Some missing works on representation learning using/within diffusion models should be included [5, 6, 7, 8, 9] in related works.`
>
>  We thank the reviewer for pointing out these valuable additional references [5–9]. We have incorporated and discussed them in the Related Works section of the revised version.

---

### Author Response · Authors · 2025-12-01

We thank the AC and reviewers for their time and their careful and constructive feedback. We are glad that reviewers find **Dispersive Loss** conceptually appealing and simple (i7Q5, 5V25, btvs, D22i), empirically effective in improving generative quality (i7Q5, 5V25, D22i), and broadly applicable across baselines and datasets (i7Q5, 5V25). Reviewers also found the paper clearly written and appreciated that our empirical results are based on strong diffusion baselines.
Across reviews, the main concerns centered on
1. the **theoretical motivation** of Dispersive Loss, including its connection to entropy-like objectives, mutual information, and contrastive/self-supervised learning (i7Q5, D22i);
2. the **relationship to prior representation-based methods**, especially REPA and related alignment/regularization approaches (D22i);
3. the **breadth and depth of empirical validation**, particularly the impact on representation quality (beyond FID), the generality across models/datasets, and the robustness of the reported gains (i7Q5, 5V25, btvs, D22i).

We believe that we have addressed these concerns through additional clarifications and experiments, together with our per-reviewer replies.

---

## 1. Clarifications
### 1.1. Scope and goals of Dispersive Loss.
Some reviewers asked whether Dispersive Loss is intended as a new representation-learning objective, a generative modeling objective, or both, and how it should be positioned relative to existing contrastive or self-supervised methods. We clarify that our primary goal is to propose a **minimal, plug-and-play representation regularizer** that can be added to existing diffusion-style (and more generally, deep generative) training pipelines:
- It is **self-contained**: it only accesses the model’s own intermediate representations and loss signals, without requiring architectural changes, extra networks, or additional data.
- It is **not** meant to replace large-scale contrastive or supervised representation learning; rather, it offers a **low-overhead regularizer** that makes the representations learned by generative models “less collapsed” and more useful, with essentially no change to the sampling procedure.

### 1.2. Theoretical motivation: entropy, mutual information, and geometry.
Reviewers requested a clearer theoretical story connecting Dispersive Loss to **entropy-like terms and mutual information**, beyond the geometric “repulsion in feature space” intuition. In the revision, we:
- Explicitly interpret Dispersive Loss as encouraging **higher entropy in the feature distribution**, i.e., spreading out representations \(Z\) rather than allowing them to collapse into a few dominant directions.
- Argue that this can be viewed as implicitly increasing **mutual information** \(I(Z;X)\): if features are more dispersed and respond sensitively to changes in the input, they carry more information about \(X\), in analogy with classical information-theoretic treatments of representation learning.
- Clarify how this perspective complements the **geometric repulsion view**: decorrelating and “pushing apart” representations across samples discourages redundancy and collapse, which is beneficial for both generation and downstream linear probes.
We present these as **intuitive, conceptual guarantees** rather than fully formal theorems.
### 1.3. Relation to REPA and other representation-based methods.
Reviewer D22i asked for a clearer comparison with **REPA** and similar representation-based regularization methods, and for justification of our “minimalist, self-contained” positioning. In the revision, we:
- Contrast Dispersive Loss with REPA-style approaches, which typically rely on **external encoders, pre-training, or additional data** and often introduce substantial extra complexity or computational overhead.
- Emphasize that Dispersive Loss is **encoder-free and data-free**: it operates purely on features from the generative model itself, and can be “dropped in” without modifying the architecture or adding new modules.
- Clarify that our goal is not to outperform REPA on its own benchmarks, but to provide a **practical regularizer** that can be widely adopted because of its simplicity and compatibility with standard generative training.

---

> ### Author Response · Authors · 2025-12-01
>
> ### 1.4. Generality across architectures and datasets.
> Reviewers also asked whether Dispersive Loss is specific to a particular backbone or dataset. In response, we clarified that our experiments include:
> - Multiple **architectures**, such as SiT-style, DiT-style, and U-Net–style models;
> - Multiple **image datasets**, including both large-scale (e.g., ImageNet) and smaller benchmarks (e.g., CIFAR-10).
>
> Under comparable training settings, Dispersive Loss consistently provides **improvements in FID** across these architectures and datasets. This supports our claim that Dispersive Loss is a **general-purpose regularizer** rather than a one-off trick for a specific model.
>
> ### 1.5. Ablations and hyperparameter robustness.
> Reviewers were interested in how sensitive our method is to the choice of hyperparameters. We therefore emphasized:
> - Ablations over the **weight of Dispersive Loss** and the choice of **which layers** to regularize
> Across these ablations, we find that Dispersive Loss is **robust over a reasonable range of settings** and does not require delicate tuning to yield improvements.
> ---
> ## 2. Additional experiments
> *(please see per-reviewer responses for more reviewer-specific experiments)*
> ### 2.1. Representation probes and linear evaluation.
> To directly address concerns about **representation quality**, we added **linear probing experiments** on intermediate features of our models. In these experiments:
> - We freeze generative backbones trained **with** and **without** Dispersive Loss, and train a linear classifier on the resulting features.
> - We observe **improvements in top-1 accuracy** when Dispersive Loss is used during training.
>
> These results support our claim that Dispersive Loss not only improves generative metrics such as FID but also produces **more discriminative internal representations**.
>
> ### 2.2. Beyond images: text translation
> To demonstrate that Dispersive Loss is not limited to vision-only settings or FID, we added a **text translation experiment** (e.g., T5-based sequence-to-sequence). In this setup:
> - We apply Dispersive Loss to intermediate sequence representations;
> - We observe a **increase in BLEU score** compared to the baseline model trained without Dispersive Loss.
> This experiment shows that the same regularizer can benefit **non-visual metrics**, reinforcing our claims about breadth of applicability.
> ### 2.3 Additional Metrics
> We also added **additional metrics and training-dynamics plots**, including:
> - **Qualitative comparison** between images generated with and without Dispersive Loss.
> - **Loss curves** that illustrate how the regularizer prevents excessive collapse of representations.
> These plots help visualize how Dispersive Loss influences training dynamics and support the connection between theoretical intuition and empirical behavior.
> ---
> We have incorporated these changes in our revision. We are thankful for the opportunity to clarify and improve the paper.

---

### Meta-Review · Area_Chair_z8j4 · 2026-01-06

**Summary:**

This paper proposes a regularizer for diffusion models which encourages internal representations to disperse. Reviewers found the method sensible and appreciated the positive experimental results, but also highlighted a lack of justification for the method, a lack of experiments comparing against diffusion-based representation learning baselines, and the modesty of improvements. In response, the authors provided a simple experiment comparing their method against a vanilla diffusion model through linear probing, and highlighted that the main objective of their method is to improve generation quality, not do representation learning.

Overall, I believe that the reviewers' concerns are reasonable: even if the main goal is to improve generation quality, if the regularizer is meant to improved representation learning, then the performance at representation learning should at least be benchmarked more thoroughly. I do not believe this concern was addressed well enough in the rebuttal. I also believe that the concern that the improvements were too modest remains. I thus recommend rejection.

**Reviewer Concerns:**

I do not believe that comparisons against diffusion-based representation learning were sufficiently addressed, nor was the modesty of some of the results.

**Reviewer Scores:**

I do not believe reviewers would have changed their scores.

---

### Decision · Program_Chairs · 2026-01-26

Reject